# Entomophages of the Colorado Potato Beetle, Population Dynamics of *Perillus bioculatus* Fabr. and Its Compatibility with Biological and Chemical Insecticides

Irina Agasyeva, Mariya Nefedova, Vladimir Ismailov and Anton Nastasy *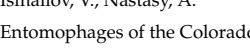

Federal Research Center of Biological Plant Protection, 350039 Krasnodar, Russia; agasieva5@yandex.ru (I.A.); dollkasneba@yandex.ru (M.N.); vlyaism@yandex.ru (V.I.)
* Correspondence: nastasy.anton@yandex.ru; Tel.: +7-(861)-228-17-76

**Abstract:** Modern plant biosecurity strategies include the use of a wide range of living organisms regulating the number, development and spread of harmful organisms at an economically safe level. We aimed to study the species composition of entomophages of the *Leptinotarsa decemlineata* Say and their efficiency in pest control. We observed the population dynamics of the stink bug *Perillus bioculatus* Fabr. and the Colorado potato beetle from 2013 to 2015. Besides, the species composition of entomophages of the Colorado potato beetle in Krasnodar Krai (a region in southwestern Russia) was researched in 2013–2015. The study showed that with a ratio of *P. bioculatus*: Colorado beetle 1:10–1:15, the efficiency of the entomophage is about 98%. In addition to *P. bioculatus*, there are other insects that feed on the Colorado potato beetle. Among them, it is worth noting *Zicrona caerulea* L., *Polistes gallicus* L., representatives of the Coccinellidae family, etc. The food base and parasitic activity of scelionid ovi-eaters and phasia flies are the main biotic factors influencing the number of predatory bugs. The possibility of combined use of *P. bioculatus* and preparations of biological origin was studied. The survival rate of adult *P. bioculatus* under the use of insecticides based on *Bacillus thuringiensis* var. *thuringiensis* and *Streptomyces avermitilis* (ex., Burg et al.) Kim and Goodfellow was 97% and 91%; that of older nymphs—58% and 52%, respectively. Chemical preparations destroyed all age stages of the predator.

**Keywords:** entomophages; predatory bug; *Perillus bioculatus* Fabr.; *Leptinotarsa decemlineata* Say; biological insecticides

## 1. Introduction

Currently, scientists are solving problems associated with the negative impact of synthetic pesticides on the quality of the environment and human health [1].

The Colorado potato beetle (CPB) (*Leptinotarsa decemlineata* Say) is a major potato pest globally causing severe economic damage. Despite a number of successful methods of biological control based on entomophages [2], nematodes [3,4], fungal and bacterial preparations [5], chemical pesticides are still the main means of crop protection. However, the CPB has an ability to rapidly develop resistance to many major classes of chemical insecticides [6–8]. There is also a danger of developing resistance of the CPB to *Bacillus thuringiensis* (Bt) toxins [9–11]. The pest resistance significantly complicates the insect control [12]. Negative environmental impacts of pesticides (including the death of nontarget species of organisms) and the desire to obtain high-quality food products free from pesticide residues predetermine the search for alternative ways to control the pest population. The use of natural enemies of the CPB appears to be the most promising one.

Biological agents in the CPB management have been applied broadly since this pest has spread widely in the United States and Canada, and then in Europe and Asia. Research was carried out in three directions: (1) the search and study of entomopathogenic microorganisms and the development of microbiological preparations based on them, (2) the study

of local entomofauna to identify the most effective parasites and predators of the pest, (3) the introduction of entomophages of the CPB from the North American continent [13,14]. Entomologists from a number of European countries have introduced natural predators to manage the CPB: predatory bugs of the subfamily Asopinae *Perillus bioculatus* Fabr., *Podisus maculiventris* Say, *Oplomus nigripennis* var. Pulcher (Dull.) and parasitic flies of the genus Doryphorophaga [15,16]. A large amount of experimental material has been accumulated on the biology of introduced predators, and studies have been carried out with the aim of acclimatizing them in the European territory [13,17]. However, none of the predators could be considered acclimatized until a population of the North American predatory *P. bioculatus* was discovered in Krasnodar Krai (Russia) in 2008 [18,19].

In 2008, when examining the common ragweed (*Ambrosia artemisiifolia* L.), employees of the Laboratory of the State Collection of Entomoacariphages and Primary Evaluation of Biological Plant Protection Products (FSBSI FRCBPP) found numerous *P. bioculatus* larvae (from 10 to 20 ind./m$^2$), actively feeding on herbiphages. This aroused great interest in the predator as a regulator of the number of the Colorado potato beetle. *P. bioculatus*, apparently, has independently acclimatized and spread in the agroecosystems of the South of Russia.

To date, more than 290 species of arthropods are known to be entomophages of the CPB, the vast majority of which are predators (94.1%), and only 5.9% are parasitic insects [17].

The role of species diversity of insects (if most of it is zoophages) is that it performs the function of a biological barrier, preventing the reproduction of pests. Having data on the diversity of insect species, it is possible to find ways to use the powerful forces of natural regulation and, above all, the activity of natural populations of entomophages [20]. Groups of predatory insects are considered one of the main regulators of the phytophage pests of agricultural crops. Therefore, the study of the species composition of insects in the agrocenosis of nightshade crops in Krasnodar Krai is of great practical importance. It helps in establishing the role of predators as components of agrobiocenoses in the regulation of the number of phytophages.

Annually, the number of pests of agricultural crops is noticeably reduced due to the predatory, parasitic insects and microorganisms. Great importance is attached to the identification and preservation of them in the fields and orchards of all land users. The use of natural factors and, above all, natural populations of entomophages limits the activity of harmful species and ensures the "co-existence" of agricultural crops with pests at a level at which they do not cause an economically tangible decrease in yield [21].

The basis of the modern concept of biological pest suppression is the management of the biological resources of agricultural landscape, conservation, activation and reproduction of natural populations of entomophages, entomopathogenic organisms and other bioagents.

The study of trophic relationships and dynamics of the number of entomophages of certain biological groups is of great practical importance for biological control. To build a technology for the biological protection of nightshade crops, it is necessary to have data on the species composition of insects dwelling in the agrocenosis of nightshade crops and take into account their useful activity. For these purposes, we studied the species composition of entomophages.

The beneficial predatory stink bug *P. bioculatus* in the potato agrocenosis as well as treatment with biological preparations will allow for controlling the CPB. Reducing the pesticide load on agrocenosis promotes the activity of native species of entomophages of the CPB. This, in turn, will reduce the number and frequency of protective measures for pest control.

The purpose of this work was to determine the species composition of the Colorado potato beetle entomophagous insects on potato culture, to study the dynamics of the number of the pest and predatory bug—*P. bioculatus*, and to study the sensitivity of *P. bioculatus* to biological and chemical insecticides.

## 2. Materials and Methods

Experiments and observations were carried out in Krasnodar Krai. Krasnodar Krai is located on the territory of the North Caucasus, in its western part, between $43°30'$ and $46°50'$ north latitude and $36°30'–41°45'$ east longitude. The total area is 76,000 km$^2$. Krai stretches for 378 km from north to south, and for 300 km from east to west.

The study of the species composition of entomophages of potato agrocenosis took place for three years (from 2013 to 2015) in areas where chemicals were not used: (1) experimental fields of the FSBSI FRCBPP (Krasnodar, $45°02'$ N $38°59'$ E) on an area of 1 ha, (2) organic farm in the village of Moldavanovskoye, Crimean district, Krasnodar Krai ($44°56'37''$ N $37°52'11''$ E) on an area of 3 ha. When studying the species composition, the methods of visual accounting and sweeping with an entomological net were used. Visual accounting was carried out by examining potato plants from the upper and lower sides of the leaves; on average, the survey lasted 10–20 s per plant. Sweeping was carried out with a standard entomological net with a 110 cm long handle, 35 cm in diameter, with 25 strokes (corresponds to 12 m$^2$) of 6 repetitions. Species were identified using the key [22] and comparative entomological collections of the FSBSI FRCBPP.

The frequency of insect occurrence was calculated using the Dajoz formula.

$$F\ (\%) = 100 \times (Pi/P), \tag{1}$$

where:

Pi is the species that was discovered;
P is the total number of examined insects.

According to the frequency of occurrence, three groups of insects were distinguished: rarely—F < 5% (+); periodically (++)—5% ≤ F < 25%; often (+++)—25% << F < 50%.

The study of the population dynamics of the predatory stink bug *P. bioculatus* and the Colorado potato beetle was carried out in the experimental plot of the FSBSI FRCBPP. The number of the CPB (eggs, larvae, adults) was determined on 50 bushes (10 samples of 5 plants) placed diagonally on the field in terms of 1 m$^2$. At the same time, the number of *P. bioculatus* (nymphs and adults) was counted. Visual records of the presence of the ragweed leaf beetle (*Zygograma suturalis* Fabr., Coleoptera, Chrysomelidae) as an additional food source for *P. bioculatus* were also carried out. The records were carried out every three days.

Studies to determine the compatibility of chemical and low-hazard preparations with *P. bioculatus* in the field were conducted on experimental plots of the FSBSI FRCBPP. Each experimental plot was 50 m$^2$, and they were located randomly. The treatment was carried out in the presence of 10% of plants inhabited by larvae with a population of 10–20 individuals of the potato leaf beetle per bush in the budding phase. After treatment, the number of surviving larvae on 10 potato plants in each variant was counted. The experiments were set up in triplicate.

In the experiments, insecticides recommended for the CPB management were used: Confidor, WSC (water-soluble concentrate—WSC containing 200 g/L of imidacloprid) (Bayer CropScience, Leverkusen, Germany), application rate—0.1 L/ha; Decis Expert, EC (emulsion concentrate—EC, contains deltramethrin 100 g/L) ("Bayer CropScience", Leverkusen, Germany) with an application rate of 0.1 L/ha; Dursban, EC (concentrate emulsion—EC, active substance-chlorpyrifos 480 g/L) ("Syngenta AG", Basel, Switzerland) application rate—2.0 g/L; Aktara, WDG (water-dispersible granules—WDG; thiamethoxam 250 g/kg) (Syngenta AG, Basel, Switzerland) with an application rate of 0.06 kg/ha. To determine the sensitivity of the entomophage to low-hazard plant protection products, we used Fitoverm, CE recommended against the CPB (emulsion concentrate—EC, Aversectin C 2 g/L) ("Pharmbiomed", Moscow, Russia) with an application rate of 0.4 L/ha and *Bitoxibacillin, P* (powder—P, *Bacillus thuringiensis* var. thuringiensis BA-1500 EA/mg, titer not less than 20 billion spores/g) (Sibbiopharm, Novosibirsk, Russia) with an application rate of 3 kg/ha. The experimental plots with entomophages and the CPB beetle were treated with the working solution of the biological preparation using a backpack hydraulic sprayer.

The effects of the drugs were taken into account by counting live individuals of *P. bioculatus* and CPB (adult and larval stages) before and after treatment. The biological effectiveness of the treatment was calculated using the modified Henderson-Tilton's formula (1955) adjusted for the control:

$$S = 100 \% \, \frac{1}{2} \, [(E_A \, \frac{1}{2} \, C_b)/(E_b \, \frac{1}{2} \, C_a)] \quad (2)$$

where:

S—the number of surviving insects corrected for control, %;
$E_A$—the number of insects in the experimental variant after treatment, ind.;
$C_b$—the number of insects in the control variant before the experiment, ind.;
$E_b$—the number of insects in the experimental variant before the treatment, ind.;
$C_a$—the number of insects in the control variant after the experiment, ind.

Statistical processing of the results was carried out using the computer software Statistica 13.2 by Duncan's multiple-range test and Excel for plotting charts and calculating confidence intervals.

## 3. Results

### 3.1. Species Composition of Entomophages of Potato Agrocenosis

As a result of the research, representatives of the orders Coleoptera, Hemiptera, Diptera, and Hymenoptera were identified. They may play an important role in the CPB management (Table 1).

**Table 1.** Species Composition of Entomophagous Insects of Potato Agrocenosis in the Central Zone of Krasnodar Krai.

| No. | Names of Species Belonging to Orders and Families | Feeding Specialization | Occurrence |
|---|---|---|---|
| | Order Coleoptera | | |
| | Family Cantharididae | | |
| 1. | *Cantharis rustica* Fallen, 1807 | P *, polyphage | ++ ** |
| | Family Coccinellidae | | |
| 2. | *Coccinella septempunctata* Linnaeus, 1758 | P, polyphage | ++ |
| 3. | *Adalia bipunctata* Linnaeus, 1758 | P, polyphage | + |
| 4. | *Harmonia quadripunctata* Pontopiddian, 1763 | P, aphids | + |
| 5. | *Harmonia axyridis* Pallas, 1773 | P, polyphage | ++ |
| 6. | *Propylea quatuordecimpunctata* Linnaeus, 1758 | P, polyphage | ++ |
| | Order Hemiptera | | |
| | Family Pyrrhocoridae | | |
| 7. | *Pyrrhocoris apterus* Linnaeus, 1758 | Phy, P, polyphage | + |
| | Family Pentatomidae | | |
| 8. | *Zicrona caerulea* Linnaeus, 1758 | P, polyphage | ++ |
| 9. | *Perillus bioculatus* Fabricius, 1775 | P, Colorado beetle | ++ |
| | Family Nabidae | | |
| 10. | *Nabis ferus* Linnaeus, 1758 | P, polyphage, small insects | + |
| | Order Diptera | | |
| | Family Syrphidae | | |
| 11. | *Sphaerophoria scripta* Linnaeus, 1758 | P, polyphage | +++ |
| | Family Tachinidae | | |
| 12. | *Ectophasia crassipennis* Fabricius, 1794 | Par * | + |

**Table 1.** *Cont.*

| No. | Names of Species Belonging to Orders and Families | Feeding Specialization | Occurrence |
|---|---|---|---|
| | Order Hymenoptera | | |
| | Family Vespidae | | |
| 13. | *Polistes gallicus* Linnaeus, 1767 | P, polyphage | + |
| | Family Scelionidae | | |
| 14. | *Trissolcus grandis* Thomson, 1861 | Par, ovarian parasite of stink bugs | ++ |
| 15. | *Trissolcus vassilievi* Mayr, 1913 | Par, ovarian parasite of stink bugs | ++ |
| 16. | *Telenomus chloropus* Thomson, 1861 | Par, ovarian parasite of stink bugs | ++ |
| | Family Formicidae | | |
| 17. | *Lasius niger* Linnaeus, 1758 | P, polyphage | ++ |
| | Order Neuroptera | | |
| | Family Chrysopidae | | |
| 18. | *Chrysoperla carnea* Stephens, 1836 | P, polyphage | + |

* P—predator; Phy—phytophage; Par—parasite; ** +—seldom; ++—periodically; +++—often.

Fourteen species of predatory and four species of parasitic insects were noted in the potato cenosis of the Central Zone of Krasnodar Krai. Specific entomophages, such as *P. bioculatus*, play a crucial part in the CPB control. However, in addition to *P. bioculatus*, the useful activity of *Zicrona caerulea* L., *Polistes gallicus* L. (Figure 1) and others has been repeatedly observed.

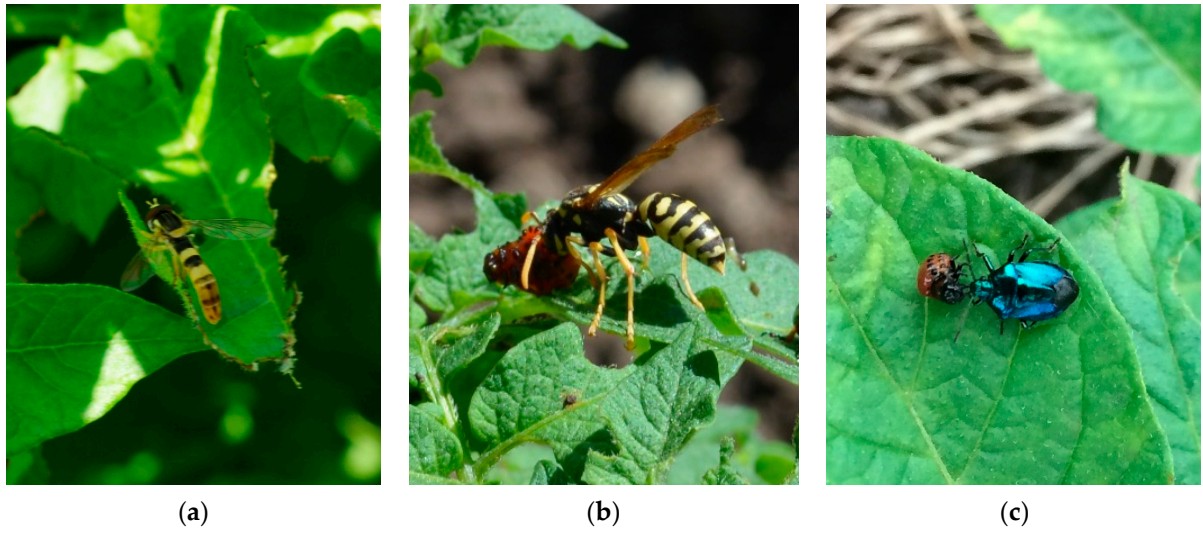

    (a)           (b)           (c)

**Figure 1.** Nonspecific entomophages of the Colorado potato beetle: *Sphaerophoria scripta* L. (**a**), *Polistes gallicus* L. (**b**), *Zicrona caerulea* L. (**c**), experimental plot of the FSBSI FRCBPP, 2014–2015.

In addition to predators, we found parasitic Hymenoptera (Table 1) capable of reducing the number of *P. bioculatus*. These include representatives of the Scelionidae family—*Trissolcus grandis* Thomson, *Trissolcus vassilievi* Mayr and *Telenomus chloropus* Thomson, as well as a fly of the Tachinidae family—*Ectophasia crassipennis* Fabr.

### 3.2. Study of the Population Dynamics of the Predatory Stink Bug Perillus bioculatus Fabr.

In our studies, we paid special attention to predatory azopine bugs, regulators of the dominant pest—the CPB *Leptinotarsa decemlineata* Say, and, primarily, to the *P. bioculatus* acclimatized in the South of Russia.

For several years, we have counted the *P. bioculatus* and its main food base—the CPB and the ragweed leaf beetle (*Zygograma suturalis* Fabr.). We have concluded that it played a decisive role in the dynamics of the number of entomophages. The regulatory role of aboriginal species of entomophages is a significant factor in reducing the number of *P. bioculatus*. In different years, there was a considerable disease of stink bug eggs (5–28%) with oviparous parasites *Trissolcus vassilievi* Mayr and *Trissolcus grandis* (Hymenoptera: Scelionidae), as well as *P. bioculatus* adults (8–15%) with phasia flies (subfamily *Phasiinae*, fam. *Tachinidae*).

During the study period (2013–2015), the most preferred food for *P. bioculatus* was eggs of the Colorado potato beetle, but it also actively fed on larvae and adults (Figure 2).

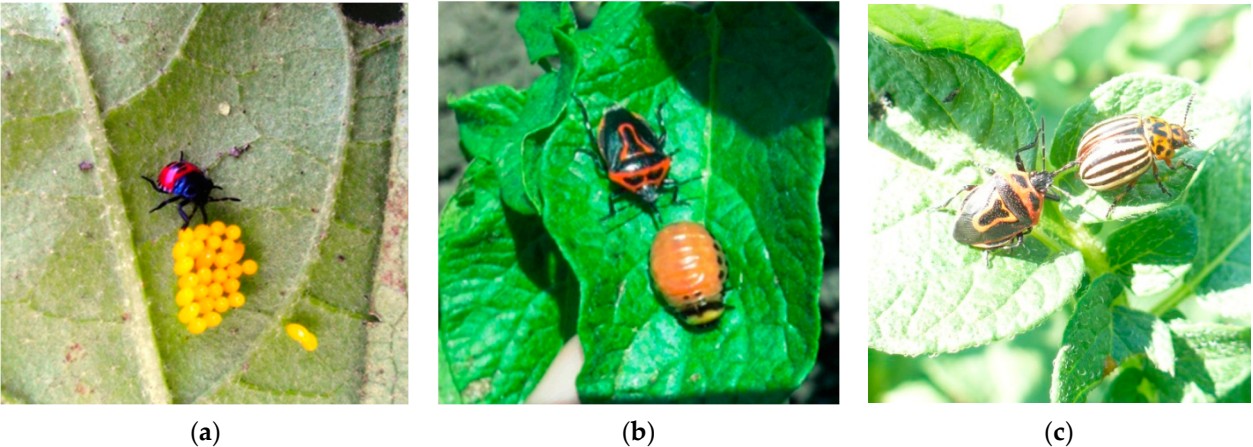

(**a**)　　　　　　　　　(**b**)　　　　　　　　　(**c**)

**Figure 2.** *P. bioculatus* feeds on eggs (**a**), larvae (**b**) and adults (**c**) of *L. decemlineata*.

The warm winter of 2013 was favorable for overwintering *P. bioculatus*. The cold and protracted spring made it possible to synchronize the development cycles of the predator and the Colorado potato beetle. The CPB appeared at the end of April; the first individuals of *P. bioculatus* were recorded around the same time.

At the beginning of May 2013, *P. bioculatus* was represented only by a red phenoform (the average ten-day temperature was about +20 °C); by mid-May, specimens with an orange coloration of the shield appeared (about +23 °C); but the white phenoform was noted only by mid-July (about +24.5 °C) (Figure 3).

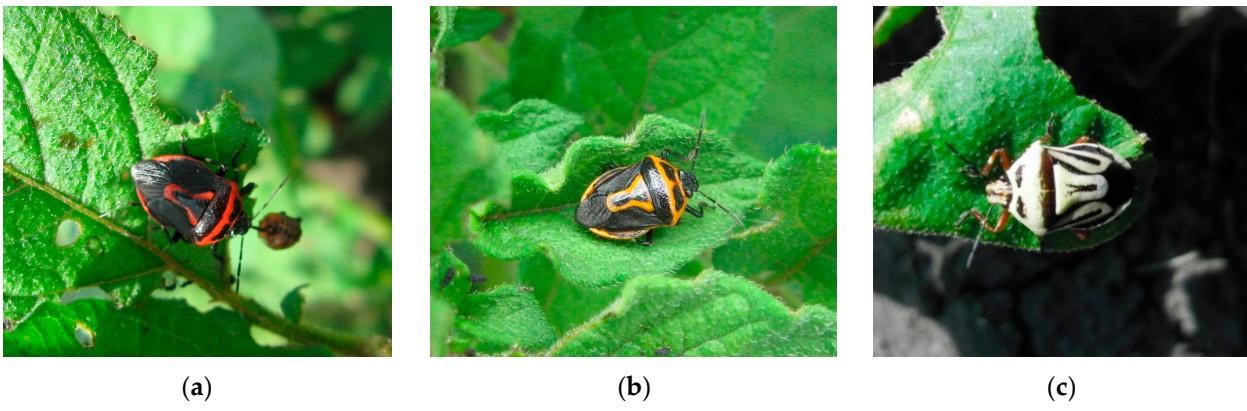

(**a**)　　　　　　　　　(**b**)　　　　　　　　　(**c**)

**Figure 3.** Red (**a**), orange (**b**), and white (**c**) *P. bioculatus* phenoform, May–July 2013.

In total, during the research period (2013–2015), insects with a red shield were noted on average about 74%, an orange one—17.7% and a white one—about 8.5%. Fent, M. and Aktac, N. (2007) reported on the cleavage of an enzyme that affects the formation of scutellum coloration at elevated temperatures [23].

In 2014, the first individuals of *P. bioculatus* were recorded on 3 May (simultaneously with the CPB overwintered generation). In the spring, warm weather set in (the temperature in the first 10 days of May exceeded the long-term average by 2.3 °C). The weather favored the CPB reproduction, the number of which significantly exceeded the ET of harmfulness. The number of the natural population of *P. bioculatus* turned out to be low and insufficient to control the pest.

Figure 4 shows the dynamics of the natural population of *P. bioculatus* and the Colorado potato beetle.

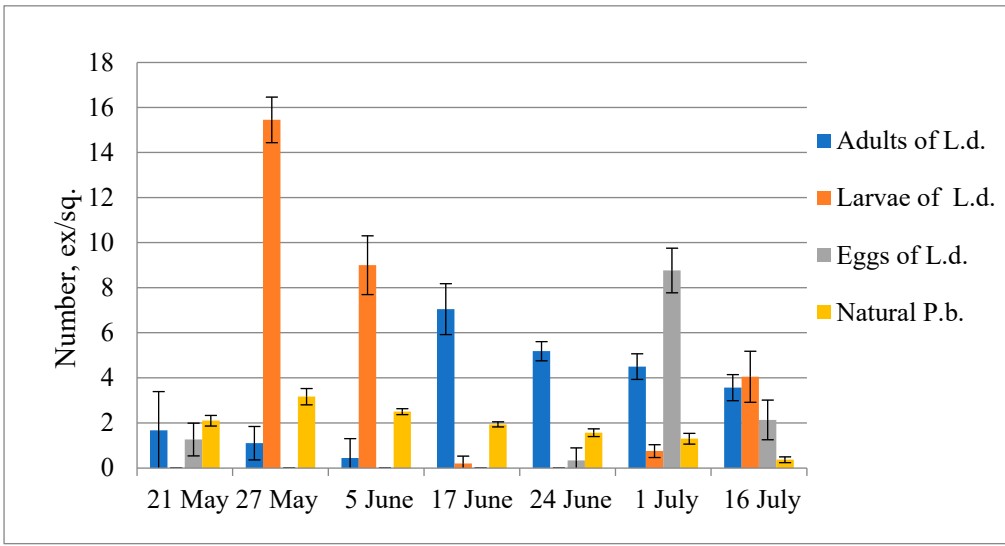

**Figure 4.** Number of adults, larvae, eggs of *L. decemlineata* (L.d.) and natural population of *P. bioculatus* (P.b.), 2014.

The number of adults and eggs of the CPB at the beginning of the last 10 days of May was about 1.5 ind./m$^2$. The number of *P. bioculatus* individuals in the natural population during this period was 2 ind./m$^2$. Owing to the presence of the CPB eggs, which are the optimal food source for the bug, the number of *P. bioculatus* increased to 3 ind./m$^2$ by the end of May. At the same time, the number of intact eggs decreased to 0 ind./m$^2$. This indicator remained until the last 10 days of June. The food of *P. bioculatus* at that time consisted of a small number of newly laid eggs (only the damaged ones found), adults and larvae of the CPB, the number of which was constantly decreasing and by mid-July amounted to 3.5–4 ind./m$^2$.

In 2015, the first individuals of *P. bioculatus* appeared in the second 10 days of May, at the end of which its abundance was 0.1 ind./m$^2$ and gradually began to increase (Figure 5). In June, the population density of the predator was high and amounted to 2.8 ind./m$^2$. However, in the first 10 days of July, the number of the entomophages began to decrease, as the CPB number dropped.

Figure 5 indicates that the number of *P. bioculatus* changes synchronously with that of the CPB, and directly depends on the number of the pest. At the end of the growing season, we observed the biological development of the entomophage on tomato and eggplant, where both the pest and the entomophage migrated after the drying of the potato leaf apparatus.

Figure 6 shows the number of eggs consumed by the predatory bug during the 2015 growing season. Thus, at the beginning of the growing season, the emergence of the harmful CPB larvae was prevented, and the potatoes were protected from the pest.

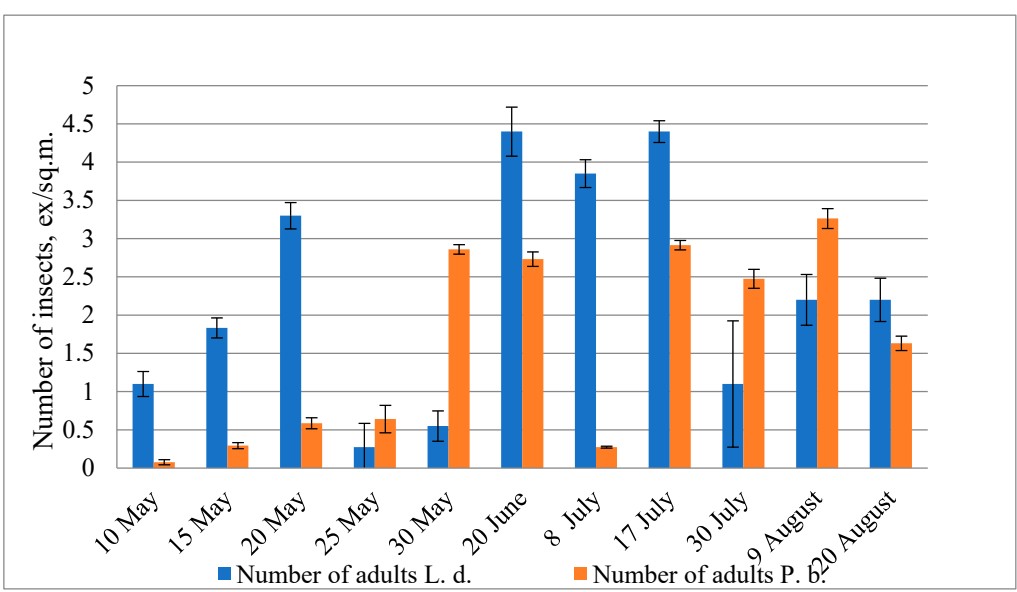

**Figure 5.** The number of *L. decemlineata* (L.d.) and *P. bioculatus* (P.b.) during the potato growing season, 2015.

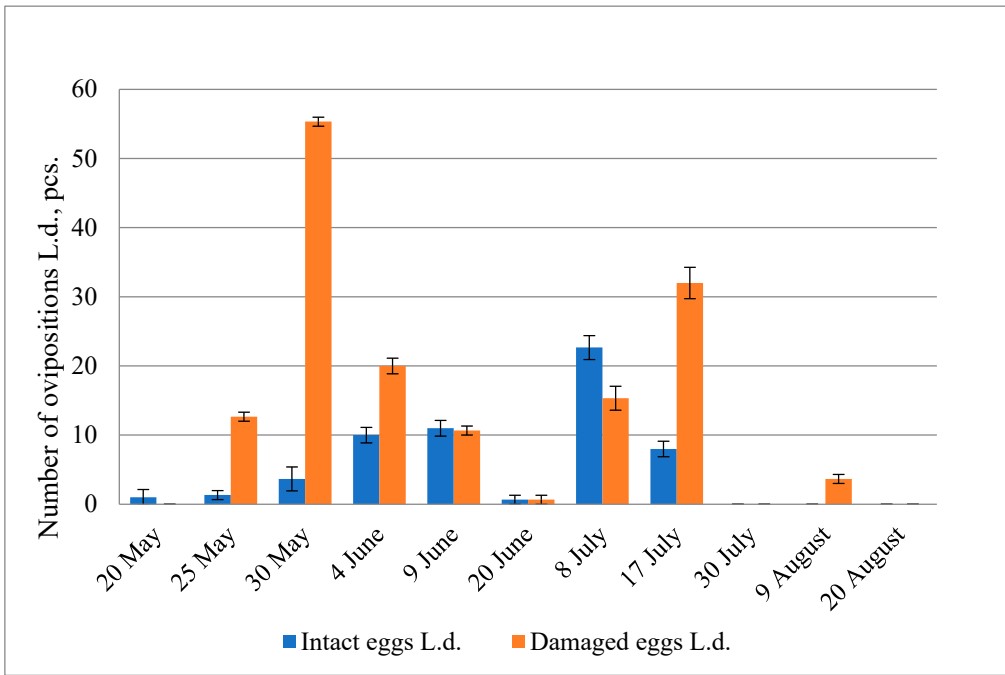

**Figure 6.** The number of intact and damaged by *P. bioculatus* (P.b.) eggs of *L. decemlineata* (L.d.), 2015.

Favorable wintering conditions for *P. bioculatus* in 2015 (early and warm spring) made it possible to synchronize the development cycles of the predator and the CPB. Thus, the biological efficiency of the natural regulation of the pest population in the experimental plots reached 95–99%. It was found that out of 54 ovipositions recorded, only one was not destroyed by the predator (98% efficiency) (Figure 7). Thus, the earliest CPB development stage (the egg stage) was suppressed, and the harmful larval stage was prevented.

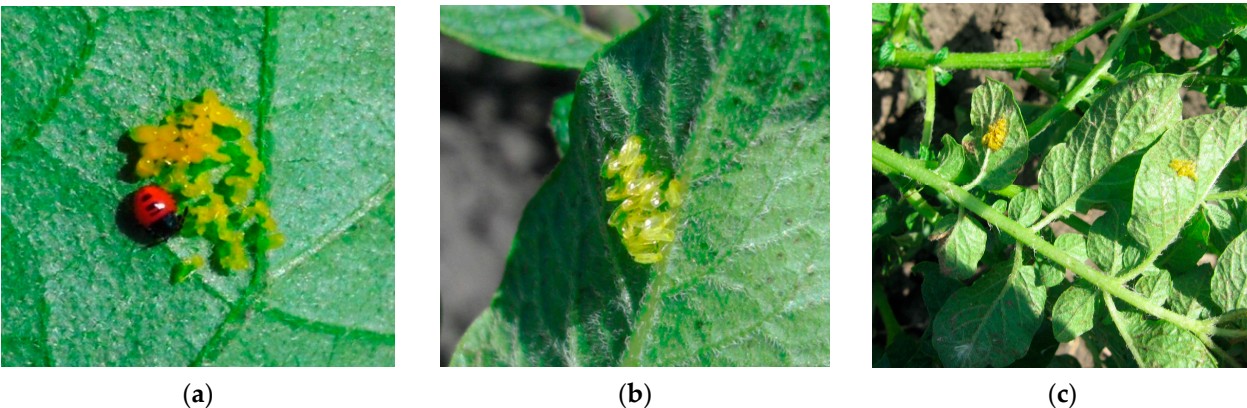

|                  (a)                  |                  (b)                  |                  (c)                  |

**Figure 7.** Biological efficiency of the *L. decemlineata* natural population regulation 2013 (**a**–**c**) eggs of *L. decemlineata* destroyed by *P. bioculatus*.

*3.3. Studying the Possibility of Joint Use of Biological and Chemical Plant Protection Products and the Predatory Bug Perillus bioculatus Fabr.*

In order to develop an effective system for the protection of nightshade crops, detailed field studies were carried out to determine the compatibility of chemical and biological preparations with the *P. bioculatus*.

Table 2 provides the results of field experiments on the compatibility of predatory bugs and low-hazard insecticides of biological origin.

**Table 2.** Sensitivity of *P. bioculatus* to Fitoverm, CE and *Bitoxibacillin, P.*

| Experience Options | Application Rate, L/ha, kg/ha | Insect Development | *P. bioculatus* Survival Rate on the 7th Day, % |
|---|---|---|---|
| Fitoverm, CE, 2 g/L | 0.4 | adults<br>older nymphs (III–IV)<br>younger nymphs (I–II) | 90.9 [b]*<br>51.5 [a]<br>0 [a] |
| *Bitoxibacillin, P*, BA-1500 EA/mg | 3 | adults<br>older nymphs (III–IV)<br>younger nymphs (I–II) | 97.0 [b]<br>57.6 [ac]<br>3 [a] |
| Aktara, WDG, 250 g/kg | 0.2 | adults<br>older nymphs (III–IV)<br>younger nymphs (I–II) | 0 [a]<br>0 [a]<br>0 [a] |
| Control | No treatment | adults<br>older nymphs (III–IV)<br>younger nymphs (I–II) | 93.9 [b]<br>87.9 [bc]<br>97.0 [b] |

* Note: there are no statistically significant differences between the options indicated by the same letter indices according to the Duncan criterion at a probability level of 95%.

We have noticed that the survival rate of adults and younger nymphs when using Fitoverm, CE was 90.9% and 0%, respectively. *Bitoxibacillin, P* did not have a significant toxic effect on the imago of the predatory bug. The survival rate of *P. bioculatus* was 97.0%; that of older nymphs—57.6%; only 3% of younger nymphs survived. Aktara, WDG completely destroyed the insects.

The study of the toxicity of chemicals for predatory bugs and their biological efficiency was carried out in the field (Table 3).

**Table 3.** Results of the field assessment of the biological efficiency of chemicals against the CPB and their toxicity to *P. bioculatus*.

| Object of Study | Variant | Application Rate, L/ha | Average Number of Adults and Larvae per Bush | | Decrease in the Number Relative to the Original, Adjusted for Control,% |
|---|---|---|---|---|---|
| | | | Before Treatment | After Treatment | |
| *Leptinotarsa decemlineata* Say. | Confidor, WSC, 200 g/L | 0.1 | 25.3 ± 3.5 | 1.7 ± 1.7 | 94.7 [ab*] |
| | Decis Expert, EC 100 g/L | 0.1 | 24.0 ± 3.4 | 12.7 ± 5.8 | 47.4 [c] |
| | Dursban, EC, 480 g/L | 2.0 | 25.7 ± 2.8 | 12.0 ± 3.0 | 53.3 [b] |
| | Aktara, WDG, 250 g/kg | 0.06 | 24.0 ± 2.3 | 0.7 ± 0.7 | 98.7 [a] |
| | Control | - | 25.3 ± 2.8 | 27.7 ± 4.3 | - |
| *Perillus bioculatus* Fabr. | Confidor, WSC, 200 g/L | 0.1 | 6.3 ± 1.7 | 0.0 ± 0.0 | 100 [a] |
| | Decis Expert, EC 100 g/L | 0.1 | 5.7 ± 1.7 | 0.0 ± 0.0 | 100 [a] |
| | Dursban, EC, 480 g/L | 2.0 | 7.3 ± 0.7 | 0.0 ± 0.0 | 100 [a] |
| | Aktara, WDG, 250 g/kg | 0.06 | 6.0 ± 1.1 | 0.0 ± 0.0 | 100 [a] |
| | Control | - | 7.0 ± 1.1 | 7.0 ± 1.1 | - |

* Note: there are no statistically significant differences between the options indicated by the same letter indices according to the Duncan criterion at a probability level of 95%.

We found that Confidor, WSC and Aktara, WDG are the most effective (95–99% biological efficiency, respectively). The effectiveness of Decis Expert, EC and Dursban, EC was low (47–53%). Thus, we assume that the pest has developed resistance to them.

We are safe to conclude that preference should be given to biological and low-risk drugs such as *Bitoxibacillin, P* and Fitoverm, CE. This will help to maintain and restore the number of entomophage populations, which, in turn, will improve the protection of nightshade crops through natural biocenotic regulation.

## 4. Discussion

At present, there is a growing understanding of the importance of biological plant protection, the use, first of all, of safe methods and means of pest control, and the production of toxic-free products [1,24,25]. The concept of the biocenotic approach underlying the biomethod is becoming more and more of a priority. A large amount of research is devoted to biodiversity conservation [26–29].

The laws of development of the productive forces of society and ecosystems underlie the functioning of agroecosystems. Entomophages and entomopathogens, being part of the composition and structure of agroecosystems, are included in the biocenotic process, i.e., in the process of functioning of populations of organisms to ensure the flow of matter, energy and information on trophic levels. Entomophages and entomopathogens, functionally participating in this process, carry out the mechanism of regulation of the number of phytophages and thus fulfill their unique regulatory and environment-forming role in agroecosystems [30]. The study of the species composition of the agrocenosis of nightshade crops in the Central Zone of Krasnodar Krai promotes the identification of the CPB natural enemies and other pests. For example, *Coccinella septempunctata* L., *Harmonia axyridis* Pallas, *Propylea quatuordecimpunctata* L. and *Nabis ferus* L. feed on eggs and larvae of the Colorado potato beetle, as well as aphids.

Under certain conditions, entomophages and other organisms can for a long time restrain a pest reproduction at an economically imperceptible level, even in cases where



the density of its population has reached or exceeded the ET (economic threshold) of harmfulness. The ecological side of the relationship between the pest and its natural enemies is reflected by the level of effectiveness of natural enemies (LENE). LENE depends on various environmental factors and, for the purpose of unification, is expressed as the ratio of the number of predator individuals to the number of prey individuals. As a result of the definition of this criterion and its practical use, it becomes possible to reduce or even completely cancel the chemical treatments of cultivated plants [31]. The ratio of *P. bioculatus* and the CPB 1:10–1:15 (up to 10 eggs and 10–15 larvae per bush) is sufficient for the almost complete elimination of the pest.

*P. bioculatus* is an introduced species. There are many positive examples of the introduction, mass breeding and use of exotic species of beneficial insects [32]. However, cases are known when the results of accidental or deliberate introduction of organisms negatively impact natural ecosystems [33]. Regarding the *P. bioculatus*, this problem refers to its aggressive behavior towards the ragweed leaf beetle, which is a useful herbiphagous. However, the ragweed leaf beetle is currently unable to influence the biological activity of the ragweed and control its numbers in Russia [34,35]. At the same time, it turned out to be an additional feeding base for the predatory bug, whose role in the economic aspect is much higher than that of the herbiphage, taking into account the loss of yield and the cost of chemical protection of potatoes.

We noted that the decrease of these phytophages in the fields of the FSBSI FRCBPP over 2–3 years led to a significant reduction in the population density of *P. bioculatus*. In 2008–2009, it reached 20–30 specimens/$m^2$; but in 2011–2015, only single individuals were recorded. Another important factor in the decline in the number of *P. bioculatus* is the regulatory role of native species of entomophages. In different years, significant infection of *P. bioculatus* eggs (5–28%) with oviparous parasites *Trissolcus vassilievi* Mayr, and *Trissolcus grandis* Thomson, *Telenomus chloropus* Thomson (Hymenoptera: Scelionidae), as well as bedbug adults (8–15%)—phasia flies (subfamily *Phasiinae*, *fam. Tachinidae*).

Often in the field, natural entomophages are unable to independently control the number of pests at an economically imperceptible level. The reason for this may be the asynchrony of the phytophage and entomophage phenology or the very low abundance of the latter after overwintering. This, in turn, contributes to the uncontrolled reproduction of harmful species. Thus, there is a need for additional releases of entomophages, the use of biological and biorational plant protection products that have no negative impact on the number of beneficial arthropods [18].

The toxic effect on beneficial organisms should be considered when selecting insecticides. In order to integrate biological plant protection products, the sensitivity of the predatory stink bug *P. bioculatus* to the biological preparations *Bitoxibacillin, P* and Fitoverm, CE was studied. We found that Fitoverm, CE at a dose of 0.4 L/ha affects the survival of nymphs of predatory bugs: for example, on the seventh day, survival in nymphs of younger ages was 0%; adults—90.9%. *Bitoxibacillin, P* at a consumption rate of 3 kg/ha had no toxic effect on the adult predatory stink bug; the survival rate of *P. bioculatus* was 97%. Preparations Confidor, WSC, Aktara, WDG, Decis Expert, CE and Dursban, CE were used as chemical standards. The use of chemicals resulted in the total destruction of the predatory stink bug.

## 5. Conclusions

We have established the decisive role of entomophages in the biocenotic regulation of the number of phytophages: a large role in reducing the number of the Colorado potato beetle is played by both the acclimatized population of the predatory stink bug *P. bioculatus*, and nonspecific entomophages, such as *Zicrona caerulea* L., *Polistes gallicus* L., *Sphaerophoria scripta* L., *Coccinella septempunctata* L., *Harmonia axyridis* Pallas, *Propylea quatuordecimpunctata* L., *Nabis ferus* L. and some others.

There is evidence that the population density of *P. bioculatus* changes synchronously with the change in the CPB population. Field experiments have shown that in Krasnodar

Krai, the ratio of the predatory stink bug *P. bioculatus* to the CPB should be 1:10–1:15 for effective pest control. Thus, LENE for the predatory stink bug *P. bioculatus* is 1:10–1:15 (for the CPB larvae of various ages—1:10, for eggs—1:15). The main biotic factors influencing the abundance of predatory bugs are the food base and the parasitic activity of scelionid ovi-eaters and phasia flies.

The possibility of the combined use of *P. bioculatus* and preparations of biological origin was studied. The survival rate of adult *P. bioculatus* when using *Bitoxibacillin, P* and Fitoverm, CE is 97% and 91%; that of older nymphs is 58% and 52%, respectively. Chemical preparations are lethal to all age stages of the predator.

Hence, we conclude the possibility of biological protection of potatoes based on the synchronization of the phenology of the Colorado potato beetle and *P. bioculatus*. This is achieved by later planting of potatoes and additional introduction of predatory bugs in the predator–prey ratio of 1:10–1:15 or treatment with effective biological preparations that do not reduce the effectiveness of entomophages.

**Author Contributions:** Conceptualization, I.A. and V.I.; methodology, I.A. and V.I.; software— statistics, A.N.; validation, I.A. and V.I.; investigation, I.A., V.I. and M.N.; writing—original draft preparation, I.A., V.I., M.N. and A.N.; writing—review and editing, A.N.; supervision, I.A. All authors have read and agreed to the published version of the manuscript.

**Funding:** The research was carried out in accordance with the State Assignment of the Ministry of Science and Higher Education of the Russian Federation within the framework of research on the topic No. FGRN-2022-0003.

**Data Availability Statement:** Not applicable.

**Conflicts of Interest:** The authors declare no conflict of interest.

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
