# Peer review of "Entomophages of the Colorado Potato Beetle, Population Dynamics of Perillus bioculatus Fabr. and Its Compatibility with Biological and Chemical Insecticides"

_agronomy, doi:10.3390/agronomy13061496_

Round 1

Reviewer 1 Report

I reviewed the manuscript entitled “Efficiency of entomophages against the Colorado Potato Beetle”.

From the title I would have expected something different: I think that with their results, through the population dynamic analysis, authors can not conclude adequately about efficiency of entomophages, althought they have shown the sysncronism of CPB and predator populations. Maybe performing a predator exclusion trials they could effectively confirm that the decrease of CPB populations was due to the increase of predator population.

Moreover, species composition of Entomophages gives information about entomophages present in those ecosystems that hypothetically can feed on colorado potato beetle. In the results authors affirm that “the useful activity of Zicrona caerulea L., Polistes gallicus L. (Figure 2) and others has been repeatedly observed.”, but they did not quantified this and did not performed, for instance, DNA based approach to identify effective CPB predators.

In general, authors also should clarify/improve/add aspects related with methodology, statistical analysis and results.

In Material and Methods authors should explain adequately all procedures. For instance, they should indicate how they performed visual accounting and collections with entomological net. How many sampling dates per year? How many sampling points per date? How many seconds/ minutes held in each visual accounting?

For the determination of the number of the CPB and P. bioculatus authors should indicate the sampling period, the number of sampling dates and the number of sampling points per date.

In insecticide trials, authors did not explain how they performed the trials. For instance, only in the results, it is possible to understand that low-hazard plant protection products were tested under laboratory conditions and chemical insecticides were testes under field conditions. Authors should indicate why they decided to evaluate the products in a different way. For laboratory trials it is important to explain, for instance, how they performed it. Did they used petri dishes or plants as support? How many replications they had? How many insets per replication? How insects were maintained? What did they feed during the seven days or until die? In the results they indicate “Laboratory-grown entomophages were released into experimental plots (Table 3).” but this is an information that should be in the material and methods.

Authors indicate that statistical analisys were performed in Statistica 13.2, but which analisys did they performed? and which variabels were analysed?

In the results authors indicate that they have introduced laboratory populations of CPB in 2013 (line 217-219) and the predator Perillus in 2014 (line 240-241). However this information, is not indicated in material and methods.

Line 133-138 – difficult to understand and we don´t have permission to access the links

Results need to be clarified and more concrete. The authors include several paragraphs of discussion and material and methods in the results section (ex: line 143-150; line 210-212; line 229-231; line 315-316). Please, include only results.

In the point 3.1 “Species composition of entomophages of potato agrocenosis” results are presented in a very general form. If authors used entomological nets, they have captured insects, they have number of insets. So they can present the real representativity of each group. Being spiders one of the most frequent predator in all ecosystems, and referred in some references as predator of CPB, I find strange that they were not observed in this study.

In the point 3.2. authors intended to “study of the population dynamics of the predatory stink bug Perillus bioculatus Fabr.” but they present a mixture of information about the predator and CPB, being difficult to follow it. Maybe they should split the information in population dynamics of predator and population dynamic of CPB, presenting figures with data of all years for each insect, and only then discuss and integrate the information. Why did you only choose the years of 2014 and 2015 to represent in the figures?

Moreover, if they introduced laboratory population they are not presenting the real dynamic population. In fact, as they did not indicate in Material and Methods how sampling was performed it seems that they are presenting punctual observations without follow an adequate methodology.

See figure 2. Legend does not coincide with figures

In the point 3.3. “Studying the possibility of joint use of biological and chemical plant protection products and the predatory bug Perillus bioculatus Fabr.” Even interesting, results are not comparable (authors performed laboratory studies for biological products and field studies for chemical products) and they should not be included in the same point, and some care should be taken when conclusions are drawn

Author Response

Dear Reviewer, thank you for your comments.

From the title I would have expected something different: I think that with their results, through the population dynamic analysis, authors can not conclude adequately about efficiency of entomophages, althought they have shown the sysncronism of CPB and predator populations. Maybe performing a predator exclusion trials they could effectively confirm that the decrease of CPB populations was due to the increase of predator population.

We have changed the title of the article. We cannot exclude the predatory bug perilus from the species composition of the entomophages, since we are not able to control the size of the natural population of the predatory bug. Therefore, it is not possible for us to conduct the research that you propose.

Moreover, species composition of Entomophages gives information about entomophages present in those ecosystems that hypothetically can feed on colorado potato beetle. In the results authors affirm that “the useful activity of Zicrona caerulea L., Polistes gallicus L. (Figure 2) and others has been repeatedly observed.”, but they did not quantified this and did not performed, for instance, DNA based approach to identify effective CPB predators.

We did not aim to conduct a DNA analysis to identify the effectiveness of predators against the Colorado potato beetle, since the predatory activity of Zicrona caerulea L., Polistes gallicus L. and others is quite widely described in the literature.

In Material and Methods authors should explain adequately all procedures. For instance, they should indicate how they performed visual accounting and collections with entomological net. How many sampling dates per year? How many sampling points per date? How many seconds/ minutes held in each visual accounting?

Materials and methods have been edited taking into account your comments.

For the determination of the number of the CPB and P. bioculatus authors should indicate the sampling period, the number of sampling dates and the number of sampling points per date.

Information on the number of repetitions has been added to materials and methods.

In insecticide trials, authors did not explain how they performed the trials. For instance, only in the results, it is possible to understand that low-hazard plant protection products were tested under laboratory conditions and chemical insecticides were testes under field conditions. Authors should indicate why they decided to evaluate the products in a different way. For laboratory trials it is important to explain, for instance, how they performed it. Did they used petri dishes or plants as support? How many replications they had? How many insets per replication? How insects were maintained? What did they feed during the seven days or until die? In the results they indicate “Laboratory-grown entomophages were released into experimental plots (Table 3).” but this is an information that should be in the material and methods.

Information on the number of repetitions has been added to materials and methodsAuthors indicate that statistical analisys were performed in Statistica 13.2, but which analisys did they performed? and which variabels were analysed?

We used the Duncan test (there is an indication of this in the note under the tables), as well as the Excel program (for plotting and calculating confidence intervals).

In the results authors indicate that they have introduced laboratory populations of CPB in 2013 (line 217-219) and the predator Perillus in 2014 (line 240-241). However this information, is not indicated in material and methods.

This study provides data only on the activity of the natural population, the mention of the laboratory population is erroneous, for which we apologize

Line 133-138 – difficult to understand and we don´t have permission to access the links

We have removed the links.Results need to be clarified and more concrete. The authors include several paragraphs of discussion and material and methods in the results section (ex: line 143-150; line 210-212; line 229-231; line 315-316). Please, include only results.

Edited these sections according to the comments

In the point 3.1 “Species composition of entomophages of potato agrocenosis” results are presented in a very general form. If authors used entomological nets, they have captured insects, they have number of insets. So they can present the real representativity of each group. Being spiders one of the most frequent predator in all ecosystems, and referred in some references as predator of CPB, I find strange that they were not observed in this study.

The study of the species composition of insects (entomophages) was carried out by different methods, and combining the results of various methods seems to us not very correct in order to represent the numbers of specific insects, so we present the data in a generalized form so that readers have an approximate idea.

Spiders are predators of many phytophages and are not an exception for potato agrocenosis, however, the purpose of our study was to list entomophages from the Insécta class, and spiders do not belong to it, although they are important in various agrocenoses. In addition, spiders can also feed on the entomophages themselves, which reduces their abundance and the efficiency of regulation of the Colorado potato beetle population.

In the point 3.2. authors intended to “study of the population dynamics of the predatory stink bug Perillus bioculatus Fabr.” but they present a mixture of information about the predator and CPB, being difficult to follow it. Maybe they should split the information in population dynamics of predator and population dynamic of CPB, presenting figures with data of all years for each insect, and only then discuss and integrate the information. Why did you only choose the years of 2014 and 2015 to represent in the figures?

This graph shows the dynamics of the number of the Colorado potato beetle, with a decrease in the number of the pest, the number of the predatory bug increases. It is not possible to separate information about the dynamics of the predator population and the dynamics of the CPB population, since we wanted to show exactly their relationship.

Academic journals usually present experimental data for 2-3 years. We chose 2014 and 2015, as they most accurately reflect the predator:prey relationship.

Moreover, if they introduced laboratory population they are not presenting the real dynamic population. In fact, as they did not indicate in Material and Methods how sampling was performed it seems that they are presenting punctual observations without follow an adequate methodology.

We ask for forgiveness for the fact that, due to our inattention, we misled. This study provides data only on the activity of the natural population, the laboratory population was not included in the potato agrocenosis in the period 2013-2015.

See figure 2. Legend does not coincide with figures

Thanks, fixed

At the point 3.3. “Studying the possibility of joint use of biological and chemical plant protection products and the predatory bug Perillus bioculatus Fabr.” Even interesting, results are not comparable (authors performed laboratory studies for biological products and field studies for chemical products) and they should not be included at the same point, and some care should be taken when conclusions are drawn

In paragraph 3.3, all studies were carried out in the field on randomized plots. We also conducted experiments in the laboratory and wanted to present them as well, but decided to leave only the results of field experiments, and forgot to make changes to the text.

Reviewer 2 Report

In general, the information and research reported in this manuscript is useful and should be published.  However, the manuscript needs some re-writing.  The Results appears to be a mixture of Introduction, Methods, Results and Discussion, and the reporting of information tends to ‘wander around’.

I have uploaded an annotated version of the manuscript with specific comments and suggestions.

There are several words that apparently did not translate correctly, for example "decades" for weeks.

Author Response

Dear reviewer, thank you very much for your comments, they are all taken into account and corrections made to the article.

Round 2

Reviewer 1 Report

The authors did a commendable job in trying to improve the manuscript. Please see some comments here and also in the attached pdf of manuscript

The new title sounds better and seems to me to better reflect the work that has been done.

However, there are still some aspects that fail.

In the methodology it is not yet fully evident what was actually done. Remember that the methodology should be described in such a way that it can be reproduced by other researchers. How many sampling dates per year? Only one? Two? In which months? For visual accounting, how many sampling points (plants observed?) were performed. For sweeping, you performed 25 strokes of how many meters or during how many second?

Line 137 and 138 – something is wrong

In line 139 when you write “  The records were carried out weekly, with three records at each site.” What is three records? Three m2?

In which concerns to statistical analisys, Duncan test is a post hoc test; before you performed an ANOVA (I THINK). Moreover you should explain which variables were included in each analysis. For the tables 2 and 3 we can see that insecticides were analysed separatly. For instance in table 2 you only include the low hazard pesticides, one negative control and Aktara, WDG, 250 g/kg that should be your positive control. on the other hand, by the table 3 we can see that you performed another analysis only with chemical insectoicides. But this is note explained in the methodology.

For table 2, how did you calculate % of survival? Counting the number of organisms before and after treatment as you present in table 3?

Why did you not performed the efficiency of low hazard pesticides against the pest? And only evaluate its impact in the predator? Only by assessing the impact on the two species will you be able to draw conclusions about which one is best to use. And you only performed this for chemical insecticides.

Concerning to the entomophage species composition, you are totally correct that combining results of different methods are not correct. I didn't suggest that. I think that you can put the data from one of the sampling methods (for example sweeping) and the remaning data (visual accounting) can be presented in the supplements. Or you can put both (without combine) in the same table. In fact, you had a lot of work but only provide a personal evaluation about occurrence. In fact it is difficult to understand what is “often” if we also don’t know how many sampling were performed.

Concerning to your comment “Spiders are predators of many phytophages and are not an exception for potato agrocenosis, however, the purpose of our study was to list entomophages from the Insécta class, and spiders do not belong to it, although they are important in various agrocenoses.” In the manuscript you don't indicate that you just want to study insects. You say in the title and in the objectives that you intend to study entomophages (organisms that feed on insects) of the CPB

Please correct the legend of table 1 – entomophages also include parasitic insects

Line 217 – Do you mean predators instead entomophages?

In the case of the predator, throughout the text authors should use the common name (stink bug) or the name of the species. (P. bioculatus). It seems to me that it is not quite correct to use perillus.

Author Response

Dear reviewer, thank you for your comments.

In the methodology it is not yet fully evident what was actually done. Remember that the methodology should be described in such a way that it can be reproduced by other researchers. How many sampling dates per year? Only one? Two? In which months? For visual accounting, how many sampling points (plants observed?) were performed. For sweeping, you performed 25 strokes of how many meters or during how many second?

Edited the methodology

Line 137 and 138 – something is wrong

Thank you for noting, fixed

In line 139 when you write “  The records were carried out weekly, with three records at each site.” What is three records? Three m2?

Fixed

In which concerns to statistical analisys, Duncan test is a post hoc test; before you performed an ANOVA (I THINK).

We consider this obvious and desided not to add this text to the article.

For table 2, how did you calculate % of survival? Counting the number of organisms before and after treatment as you present in table 3?

Yes, that's right, the methodology for testing insecticides was the same for experiments to study the possibility of combining biological and chemical insecticides with P. bioculatus.

Moreover you should explain which variables were included in each analysis. For the tables 2 and 3 we can see that insecticides were analysed separatly. For instance in table 2 you only include the low hazard pesticides, one negative control and Aktara, WDG, 250 g/kg that should be your positive control. on the other hand, by the table 3 we can see that you performed another analysis only with chemical insectoicides. But this is note explained in the methodology.

Why did you not performed the efficiency of low hazard pesticides against the pest? And only evaluate its impact in the predator? Only by assessing the impact on the two species will you be able to draw conclusions about which one is best to use. And you only performed this for chemical insecticides.

The objectives of the study included studying the effect of chemical and biological insecticides on the predatory bug P. bioculatus.

To date, many studies have been done on the territory of the Russian Federation* about effectiveness of bioinsecticides against Colorado potato beetle, therefore, from the literature data, we know that the effectiveness of biological inseticides based on Bt (Bitoxibacillin, F) is about 70%, preparations based on abamectins (Fitoverm, СE) - about 77%. Also it was important for us to show that Colorado potato beetle develops resistance to chemical insecticides and only few of the chemicals are able to control its numbers, while at the same time they kill almost all of the P. bioculatus and most likely other useful species. Therefore, we give information about dangers of chemical insecticides.

The question arises, are biological insecticides harmful for entomophages? In our work, we show that they are not as toxic as chemical ones, but some may have negative effect on larvae. All these results must be taken into account in order to create a potato protection system that allows you to save as many useful species as possible and control the number of pests.

  1. Эффективность микробиологических препаратов против основных вредителей овощных, ягодных культур и картофеля в Ленинградской области / С. А. Доброхотов, А. И. Анисимов, С. Д. Гришечкина [и др.] // Сельскохозяйственная биология. – 2015. – Т. 50, № 5. – С. 694-704. – DOI 10.15389/agrobiology.2015.5.694rus. – EDN UXSROR.
  2. Биологическая эффективность химических инсектицидов и биопрепаратов против колорадского жука на различных сортах картофеля в условиях Сибири / Н. С. Чуликова, В. П. Цветкова, Л. В. Семерикова, А. А. Малюга // Вестник защиты растений. – 2012. – № 3. – С. 50-53. – EDN PJJXFD.
  3. Бутов, А. В. Экологическое качество картофеля при биологизации высокоинтенсивной технологии его возделывания и поливе / А. В. Бутов, А. А. Мандрова // Техника и технология пищевых производств. – 2018. – Т. 48, № 2. – С. 170-177. – DOI 10.21603/2074-9414-2018-2-170-177. – EDN YWOFXN.
  4. Чуликова, Н. С. Экономическая эффективность использования инсектицидов против колорадского жука на разных сортах картофеля / Н. С. Чуликова // Вестник НГАУ (Новосибирский государственный аграрный университет). – 2014. – № 4(33). – С. 65-69. – EDN UCKVPL.
  5. Алексеева, К. Л. Новые биоинсектоакарициды против вредителей овощных культур и картофеля / К. Л. Алексеева, Р. А. Багров, Л. Г. Сметанина // Картофель и овощи. – 2023. – № 5. – С. 19-23. – DOI 10.25630/PAV.2023.57.83.002. – EDN BQLEOQ.

Concerning to the entomophage species composition, you are totally correct that combining results of different methods are not correct. I didn't suggest that. I think that you can put the data from one of the sampling methods (for example sweeping) and the remaning data (visual accounting) can be presented in the supplements. Or you can put both (without combine) in the same table. In fact, you had a lot of work but only provide a personal evaluation about occurrence. In fact it is difficult to understand what is “often” if we also don’t know how many sampling were performed.

We would like to leave just such an option for providing approximate data, however, we agree that readers need clarifications for a better understanding. We have expanded the methodology by adding formula with describtion and appropriate explanations.

Concerning to your comment “Spiders are predators of many phytophages and are not an exception for potato agrocenosis, however, the purpose of our study was to list entomophages from the Insécta class, and spiders do not belong to it, although they are important in various agrocenoses.” In the manuscript you don't indicate that you just want to study insects. You say in the title and in the objectives that you intend to study entomophages (organisms that feed on insects) of the CPB

Thank you for your comment, we have added a clarification to the goals and objectives of the article.

Please correct the legend of table 1 – entomophages also include parasitic insects

We corrected the name of the table, the legend was not changed, because we think it is important to know how entomophages act: they prey or parasitize.

Line 217 – Do you mean predators instead entomophages?

Yes, corrected.

In the case of the predator, throughout the text authors should use the common name (stink bug) or the name of the species. (P. bioculatus). It seems to me that it is not quite correct to use perillus.

Changed all perillus to P. bioculatus

Reviewer 2 Report

The revisions appear fine to me.  You may want to define your use of the word "decade" the first time it appears in the text.  In English, decade generally refers to periods of 10 years.

Author Response

Dear reviewer, thank you for your comments.

We have changed every "decade" to "ten days".